# Revisiting Self-Distillation

## Abstract

Knowledge distillation is the procedure of transferring "knowledge" from a large model (the teacher) to a more compact one (the student), often being used in the context of model compression. When both models have the same architecture, this procedure is called self-distillation. Several works have anecdotally shown that a self-distilled student can outperform the teacher on held-out data. In this study, we conduct a comprehensive analysis of self-distillation with a focus on vision classification across various settings. First, we show that even with a highly accurate teacher, self-distillation allows a student to surpass the teacher in all cases. Secondly, we revisit published works on self-distillation and provide empirical experiments that suggest potential incompleteness. Third, we provide an alternative explanation for the dynamics of self-distillation through the lens of loss landscape geometry. We conduct extensive experiments to show that self-distillation leads to flatter minima, thereby resulting in better generalization. Finally, we study what properties can self-distillation transfer from teachers to students, beyond task accuracy. We show that a student can inherit natural robustness by leveraging the soft outputs of the teacher, while merely training on ground-truth labels will make the student less robust.

## 1 Introduction

In recent years, deep neural networks have found success in various tasks such as image classification (Krizhevsky et al. (2012); Simonyan & Zisserman (2015); Dosovitskiy et al. (2021), object detection Simonyan & Zisserman (2015)), speech recognition (Baevski et al. (2020)), and language understanding (Devlin et al. (2019)). But their success comes at the cost of incurring billions of model parameters. As a consequence, it can be very challenging to deploy such cumbersome models on devices with constrained resources, and a plethora of model compression and acceleration methods have been developed to address this challenge.

One such method is knowledge distillation (KD), introduced by Bucila et al. (2006) and Hinton et al. (2015) as a method of transferring knowledge from a large model (teacher) to another lightweight model (student) that is much easier to deploy without significant loss in performance. The intuition is that during training, the model needs to sift through a large set of possibly massive, highly redundant datasets, so a vast amount of representation capacity is needed. But during inference, the learned features might well be represented using smaller models.

In the original KD setting, the student model has fewer parameters than the teacher, thereby resulting in improved efficiency. However, even if model compression is not the goal, it is now folklore that distillation leads to improved model performance. A series of recent works have explored the setting when *the teacher and student architectures are identical*. Somewhat curiously, here too, KD leads to uniform boosts in student test accuracy (Furlanello et al. (2018); Yang et al. (2018); Ahn et al. (2019); Mobahi et al. (2020); Borup & Andersen (2021); Zhang & Sabuncu (2020); Stanton et al. (2021)). This special case is often referred as *self-distillation*), and will be the central focus of our work.

Despite its promise, the reasons behind the success of self-distillation are not well-understood. At the face of it, both teacher and student have access to the same training dataset; the model capacities of the teacher and the student are identical; the training algorithm is identical (modulo possible choices of hyper-parameters). Where, then, are the benefits of self-distillation coming from?

**Our contributions.** Our goal is *not* to propose a new approach for self-distillation, or to fix issues with existing approaches. Rather, we systematically investigate the behavior of self-distillation by revisiting several existing published results, attempting to validate them, and uncovering further insights. Our specific contributions are as follows.

***Surpassing the teacher***. First, we perform a series of careful self-distillation experiments on modern image classification benchmarks. We confirm that *even* when the teacher has very high test accuracy, self-distillation can still enable the student to outperform its teacher.

***Probing the multi-view hypothesis***. Second, we revisit an existing theory of knowledge distillation called the *multi-view hypothesis*, proposed by Allen-Zhu & Li (2023). At a high-level, the hypothesis states that the teacher (for various reasons) typically only learns a strict subset of "views" (or facets) of the input data, and self-distillation enables the student to learn the rest of these views. We design a series of experiments to assess the hypothesis's potential limitations in explaining self-distillation.

***Loss landscape analysis.*** Third, we investigate self-distillation through the lens of loss landscape geometry. We conduct a series of experiments to show that self-distillation encourages the student to find flatter minima (relative to the teacher). These findings are consistent with recent theoretical results on KD for shallow (kernel) models (Mobahi et al. (2020)), and can be viewed as an alternative explanation for why self-distillation works: adding a distillation term flattens the loss landscape around minima, thereby improving generalization.

***Beyond test accuracy.*** Finally, the vast majority of work on self-distillation has focused on transferring teacher knowledge in the sense of test accuracy. What other benefits can self-distillation provide? We address the ability of self-distillation to transfer *robustness to natural distribution shifts* from teacher models to student models. Our findings suggest that, when given a robust teacher model, the student model can inherit some of the robustness, though there is a trade-off between in-distribution and out-of-distribution performance.

Overall, we confirm the intuition that self-distillation *is a useful strategy for boosting test accuracy*, although existing explanations for this intuition fall short. Further, we also establish the intriguing property that self-distillation *can transfer beneficial teacher properties beyond high test accuracy*, and therefore is worthy of more careful study.

## 2    Related Work

**Knowledge distillation.** Since its original introduction in (Bucila et al. (2006); Hinton et al. (2015)), many subsequent papers have introduced several refinements to KD. Romero et al. (2015) focus on the intermediate representations by using regression to match the teacher and student feature activations. Similarly, Zagoruyko & Komodakis (2017) deals with the feature maps instead of the output logits. Tian et al. (2020) use a contrastive-based objective for transferring knowledge between networks. Park et al. (2019) utilizes the distance-wise and angle-wise distillation losses that penalize structural differences in relations. Mishra & Marr (2018) and Polino et al. (2018) combines KD with network quantization to reduce bit precision of activations and weights. Xu et al. (2017) use a conditional adversarial network to learn a loss function for knowledge distillation. Yin et al. (2020) generate class-conditional images for data-free KD. Zeng & Martinez (2000); Ba & Caruana (2014) show that we can match the performance of an ensemble or deep neural networks by teaching the student to mimic the output of the teacher. Additionally, KD has been explored beyond supervised learning. Lopez-Paz et al. (2016) extend KD to unsupervised, semi-supervised, and multi-task learning settings by combining frameworks from Hinton et al. (2015); Vapnik & Izmailov (2015). Applications of KD have even made their way to recommender systems (Kang et al. (2021; 2020)), image retrieval (Chen et al. (2018)), federated learning (Lin et al. (2020)), and graph similarity computation (Qin et al. (2021)).

**Self-distillation.** Several attempts to explain the behavior of self-distillation have already been made. Furlanello et al. (2018) shows that "dark knowledge" is a form of importance weighting. Dong et al. (2019) demonstrates that early-stopping is essential for self-distillation to harness dark-knowledge. Zhang & Sabuncu (2020) provides empirical evidence that diversity in teacher predictions is correlated with the performance of the student in self-distillation. Based on this, they offer a new interpretation for teacher-student training as amortized a posteriori estimation of the softmax probability outputs, such that teacher predictions allow

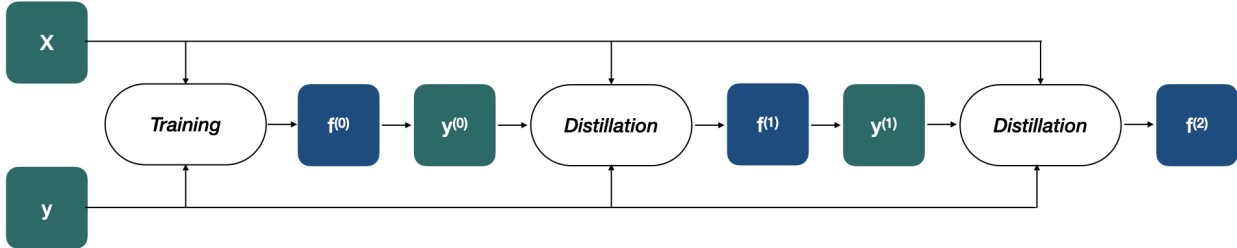

Figure 1: **Illustration of 2-round self-distillation.** $f^{(0)}$ *is the model trained from scratch using only ground-truth labels.* $f^{(1)}$ *is trained through self-distillation using both ground-truth labels $y$ and soft-labels $y^{(0)}$ from its teacher. The same procedure is used to train $f^{(2)}$, where we soft-labels from $f^{(1)}$.*

instance-specific regularization. They also propose a novel instance-specific label smoothing techniques that directly increase predictive diversity.

Mobahi et al. (2020) provide a theoretical analysis of self-distillation in the classical regression setting where the student model is only trained on the soft labels provided by the teacher. In particular, they fit a nonlinear function to training data with models belonging to a Hilbert space under $L_2$ regularization. In this setting, multi-round self-distillation is progressively limiting the number of basic functions to represent the solution. Additionally, Borup & Andersen (2021) build upon the previous analysis by also including the weighted-ground truth targets in the self-distillation procedure. They demonstrate that for fixed distillation weights, the ground-truth targets lessen the sparsification and regularization effect of the self-distilled solution. However, we note that both Mobahi et al. (2020) and Borup & Andersen (2021) use the Mean Square Error (MSE) for the objective function, and therefore their results do not directly apply to image classification models trained using the cross-entropy loss.

Allen-Zhu & Li (2023) study self-distillation under a more practical setting where the student is trained on a combination of soft-labels from the teacher and ground-truth targets. Specifically, the student objective function consists of a cross-entropy loss in the usual supervised task, and a Kullback-Leibler divergence term to encourage the student match the soft probabilities of the teacher model. They also introduce the "multi-view" hypothesis to explain how ensemble, knowledge distillation, and self-distillation work. We will discuss the hypothesis in more detail in Section 5.1. Finally, Stanton et al. (2021) systematically study the nature of (standard) knowledge distillation. They particularly study the problem through *fidelity*: how well the student can match its teacher's predictions, and *generalization*: the performance of a student on unseen held-out data. The work of Zhang et al. (2019) is perhaps closest to ours in spirit. However, their technical definition of self-distillation is different from what we consider, and therefore their observations do not directly port over to our setting.

## 3 Preliminaries

Consider the supervised setting where $\mathcal{X}$ and $\mathcal{Y}$ denote the input and output (label) space respectively with $|\mathcal{Y}| = k$. We wish to learn a classifier $f : \mathcal{X} \times \theta \to \mathbb{R}^k$ with parameters $\theta$ that maps input feature $x \in \mathcal{X}$ to a predictive distribution over $\mathcal{Y}$. Specifically, let $\mathbb{P}(y = i|\boldsymbol{x}, \theta) = \sigma_i(f(\boldsymbol{x}, \theta))$ where $\sigma(\cdot)$ is the standard softmax function. We define $f(\boldsymbol{x}, \theta)$ as the logits of the classifier $f$. We let $f_t$ and $f_s$ be respectively predictive functions representing the teacher and student, parameterized by $\theta_t$ and $\theta_s$. These functions are typically implemented as deep neural networks. When we refer to an ensemble of models, the logits $(\boldsymbol{z}_1, ..., \boldsymbol{z}_m)$ where $\boldsymbol{z}_i = f_i(x, \theta_i)$ are averaged to form the final logit vector, i.e. $\boldsymbol{z}_{ens} = \frac{1}{m} \sum_{i=1}^{m} \boldsymbol{z}_i$.

In conventional knowledge distillation, given a pre-trained teacher model, a student model is trained to emulate the teacher by minimizing the following objective:

$$\mathcal{L}_{KD} = \alpha \mathcal{L}_{CE}(\boldsymbol{z}_s, \boldsymbol{y}) + (1 - \alpha)\mathcal{L}_{KL}(\boldsymbol{z}_s, \boldsymbol{z}_t).$$

In the above equation, $\mathcal{L}_{CE}(\boldsymbol{z}_s, \boldsymbol{y}) := -\sum_{j=1}^{k} y_j \log \sigma_j(\boldsymbol{z}_s)$ is the usual cross-entropy loss between the student logits $z_s$ and labels $y$, and

$$\mathcal{L}_{KL}(\boldsymbol{z}_s, \boldsymbol{z}_t) := \tau^2 \sum_{j=1}^{k} \sigma_j(\boldsymbol{z}_t/\tau) \log \sigma_j(\boldsymbol{z}_t/\tau) - \sigma_j(\boldsymbol{z}_t/\tau) \log \sigma_j(\boldsymbol{z}_s/\tau) \, .$$

is the Kullback-Leibler divergence between the scaled student and teacher logits. Here, $\tau > 0$ is a temperature hyperparameter, $\boldsymbol{z}_t = f(\boldsymbol{x}, \theta_t)$, and $\boldsymbol{z}_s = f(\boldsymbol{x}, \theta_s)$, while $\alpha \in [0,1)$ is a constant hyperparameter that controls the relative importance of the cross-entropy and Kullback-Leibler terms.

For self-distillation, the teacher and student have the same model architecture. At round 0, the teacher model is trained from scratch. Subsequently, for every round of distillation, the teacher is the student model obtained in the previous step. We denote the model at the $n^{th}$ step of distillation as $f^{(n)}$, parameterized with $\theta_n$. See Figure 1 for an illustration.

## 4 Does The Student Always Surpass The Teacher?

First, we revisit (folklore) intuition in self-distillation, and check whether it is indeed correct.

We start by noticing that teacher model accuracies that were previously reported in related literature on self-distillation (Mobahi et al. (2020); Borup & Andersen (2021); Allen-Zhu & Li (2023)) *almost always lag behind the state-of-the-art.* See Table 1. Therefore, it could be the case that any gains by a distilled student over the teacher might have been illusory, and could have been nullified if the teacher itself was trained better.

Self-distillation is often used with the underlying assumption that the student must improve upon a teacher, and existing results on self-distillation have mostly been based on this assumption. There arises the natural question: can self-distillation *always* be expected to improve upon a teacher trained on the same dataset from scratch (i.e., using only cross entropy)? Specifically, is self-distillation a useful strategy that can improve upon even with well-trained competitive teachers? We demonstrate that this is in fact true through a series of experiments.

We know that model performance can, in practice, be further improved by (1) choosing the right set of hyperparameters and (2) adopting advanced data augmentation methodologies (Devries & Taylor (2017), Cubuk et al. (2019)). We leverage these to train better-performing teachers than ones that have been previously reported. We then train student models using self-distillation for a variety of architectures and datasets, and measure benefits (if any) of self-distillation in terms of test accuracy.

**Experiment setup.** For our experiments, we consider two architectures: ResNet18 and VGG16, trained on CIFAR-10 and CIFAR-100. We use several performance-improving heuristics, including a cosine learning rate schedule and early stopping. We also leverage modern data augmentation techniques, specifically AutoAugment (Cubuk et al. (2019)) and Cutout (Devries & Taylor (2017)), to train more accurate models. We choose the best models in every setting, and then use self-distillation to train the corresponding student models. We report all performance numbers in Table 1.

Table 1 compares the reported teachers' test accuracy on the CIFAR-10/100 dataset to the student models we trained via self-distillation (details of training hyperparameters are provided in the Appendix). We infer the following observations based on Table 1:

1. Teacher models used in Mobahi et al. (2020), Borup & Andersen (2021), and Allen-Zhu & Li (2023) are relatively weak baselines, showing that in principle any students distilled using these teachers could be obtaining performance boosts simply by virtue of better training.

2. In contrast, our ResNet18 teacher model achieves 95.56% test accuracy, which is even higher than larger architectures (e.g. ResNet34, ResNet50) used in previously published work on self-distillation; see the bottom several rows on Table 1.

3. However, self-distillation does indeed boost generalization (e.g., 97.16% → 97.40%) even when the teacher is a strong classifier trained with heavy-duty data augmentation.

Table 1: **Comparison of reported teacher and student performances from published self-distillation literature.** *A proper choice of training hyperparameters makes a baseline teacher outperform the self-distilled students reported in Mobahi et al. (2020), Borup & Andersen (2021), and Allen-Zhu & Li (2023). Moreover, our choice of architecture (e.g., ResNet18) has fewer parameters than the models of Mobahi et al. (2020); Borup & Andersen (2021); Allen-Zhu & Li (2023). However, self-distillation does improve the generalization when the teacher is trained with advanced data augmentation techniques such as Cutout (Devries & Taylor (2017)) and AutoAugment (Cubuk et al. (2019)).*

| Literature | Architecture | Dataset | Teacher | Student |
|---|---|---|---|---|
| Mobahi et al. (2020) | ResNet50 | CIFAR-10 | 80.5% | 81.3% |
| Mobahi et al. (2020) | VGG16 | CIFAR-100 | 55.0% | 56.5% |
| Borup & Andersen (2021) | ResNet34 | CIFAR-10 | 84% | 85% |
| Allen-Zhu & Li (2023) | ResNet34 | CIFAR-10 | 93.65% | 94.21% |
| Allen-Zhu & Li (2023) | ResNet34 | CIFAR-100 | 71.66% | 73.14% |
| Ours | ResNet18 | CIFAR-10 | **95.56**% | **95.84**% |
| Ours + Data Aug. (Devries & Taylor (2017); Cubuk et al. (2019)) | ResNet18 | CIFAR-10 | **97.16**% | **97.40**% |
| Ours | VGG16 | CIFAR-10 | **94.39**% | **94.50**% |
| Ours + Data Aug. (Devries & Taylor (2017); Cubuk et al. (2019)) | VGG16 | CIFAR-10 | **96.19**% | **96.49**% |
| Ours | ResNet18 | CIFAR-100 | **76.30**% | **77.73**% |
| Ours + Data Aug. (Devries & Taylor (2017); Cubuk et al. (2019)) | ResNet18 | CIFAR-100 | **78.22**% | **80.71**% |
| Ours + Data Aug. (Devries & Taylor (2017); Cubuk et al. (2019)) | ViT-S/32 | CIFAR-10 | **98.40**% | **98.46**% |
| Ours + Data Aug. (Devries & Taylor (2017); Cubuk et al. (2019)) | ViT-S/32 | CIFAR-100 | **90.39**% | **90.41**% |
| Ours + Data Aug. (Devries & Taylor (2017); Cubuk et al. (2019)) | ViT-B/32 | CIFAR-10 | **98.96**% | **98.98**% |
| Ours + Data Aug. (Devries & Taylor (2017); Cubuk et al. (2019)) | ViT-B/32 | CIFAR-100 | **92.96**% | **93.02**% |

Table 2: **Self-distillation results on CIFAR-10.** *Data augmentation means leveraging Cutout and AutoAugment techniques. We report mean and standard deviations of test accuracy from three independent runs. ↑ (resp. ↓) stands for the increase (resp. decrease) in test accuracy relative to its teacher.*

| Architecture | Dataset | Data Aug. | $\alpha$ | Teacher | Round 1 | Round 2 | Round 3 | SAM |
|---|---|---|---|---|---|---|---|---|
| ResNet18 | CIFAR-10 | No | 0.2 | $95.57 \pm 0.15$ | $95.80 \pm 0.05$(↑) | $95.58 \pm 0.13$(↓) | $95.62 \pm 0.09$(↑) | $96.25 \pm 0.06$ |
| ResNet18 | CIFAR-10 | No | 0.5 | $95.57 \pm 0.15$ | $95.84 \pm 0.10$(↑) | $95.60 \pm 0.17$(↓) | $95.59 \pm 0.01$(↓) | $96.25 \pm 0.06$ |
| ResNet18 | CIFAR-10 | No | 0.8 | $95.57 \pm 0.15$ | $95.74 \pm 0.09$(↑) | $95.55 \pm 0.10$(↓) | $95.62 \pm 0.09$(↑) | $96.25 \pm 0.06$ |
| ResNet18 | CIFAR-10 | Yes | 0.2 | $97.15 \pm 0.07$ | $97.24 \pm 0.05$(↑) | $97.39 \pm 0.01$(↑) | $97.44 \pm 0.04$(↑) | $97.42 \pm 0.04$ |
| ResNet18 | CIFAR-10 | Yes | 0.5 | $97.15 \pm 0.07$ | $97.40 \pm 0.04$(↑) | $97.36 \pm 0.05$(↓) | $97.38 \pm 0.04$(↑) | $97.42 \pm 0.04$ |
| ResNet18 | CIFAR-10 | Yes | 0.8 | $97.15 \pm 0.07$ | $97.28 \pm 0.07$(↑) | $97.38 \pm 0.11$(↑) | $97.43 \pm 0.05$(↑) | $97.42 \pm 0.04$ |
| VGG16 | CIFAR-10 | No | 0.2 | $94.39 \pm 0.11$ | $94.45 \pm 0.12$(↑) | $94.25 \pm 0.09$(↓) | $94.25 \pm 0.04$(−) | $95.02 \pm 0.17$ |
| VGG16 | CIFAR-10 | No | 0.5 | $94.39 \pm 0.11$ | $94.50 \pm 0.12$(↑) | $94.26 \pm 0.14$(↓) | $94.16 \pm 0.17$(↓) | $95.02 \pm 0.17$ |
| VGG16 | CIFAR-10 | No | 0.8 | $94.39 \pm 0.11$ | $94.38 \pm 0.07$(↓) | $94.35 \pm 0.06$(↓) | $94.30 \pm 0.11$(↓) | $95.02 \pm 0.17$ |
| VGG16 | CIFAR-10 | Yes | 0.2 | $96.19 \pm 0.05$ | $96.36 \pm 0.15$(↑) | $96.33 \pm 0.05$(↓) | $96.29 \pm 0.05$(↓) | $96.61 \pm 0.12$ |
| VGG16 | CIFAR-10 | Yes | 0.5 | $96.19 \pm 0.05$ | $96.49 \pm 0.08$(↑) | $96.36 \pm 0.08$(↓) | $96.39 \pm 0.04$(↑) | $96.61 \pm 0.12$ |
| VGG16 | CIFAR-10 | Yes | 0.8 | $96.19 \pm 0.05$ | $96.36 \pm 0.03$(↑) | $96.42 \pm 0.06$(↑) | $96.37 \pm 0.05$(↓) | $96.61 \pm 0.12$ |
| ViT-S/32 | CIFAR-10 | Yes | 0.2 | $98.40 \pm 0.00$ | $98.46 \pm 0.04$(↑) | $98.46 \pm 0.04$(−) | $98.43 \pm 0.02$(↓) | $98.52 \pm 0.03$ |
| ViT-S/32 | CIFAR-10 | Yes | 0.5 | $98.40 \pm 0.00$ | $98.46 \pm 0.01$(↑) | $98.49 \pm 0.02$(↑) | $98.45 \pm 0.02$(↓) | $98.52 \pm 0.03$ |
| ViT-S/32 | CIFAR-10 | Yes | 0.8 | $98.40 \pm 0.00$ | $98.48 \pm 0.02$(↑) | $98.46 \pm 0.04$(↓) | $98.48 \pm 0.02$(↑) | $98.52 \pm 0.03$ |
| ViT-B/32 | CIFAR-10 | Yes | 0.2 | $98.96 \pm 0.01$ | $98.98 \pm 0.03$(↑) | $98.98 \pm 0.01$(−) | $98.99 \pm 0.01$(↑) | $99.02 \pm 0.01$ |
| ViT-B/32 | CIFAR-10 | Yes | 0.5 | $98.96 \pm 0.01$ | $99.00 \pm 0.02$(↑) | $98.98 \pm 0.01$(↓) | $98.99 \pm 0.04$(↑) | $99.02 \pm 0.01$ |
| ViT-B/32 | CIFAR-10 | Yes | 0.8 | $98.96 \pm 0.01$ | $99.01 \pm 0.01$(↑) | $98.99 \pm 0.01$(↓) | $99.01 \pm 0.01$(↑) | $99.02 \pm 0.01$ |

We therefore conclude that the aforementioned published results on self-distillation are directionally correct: self-distillation really does improve upon teacher accuracy, even when the teachers themselves are strong classifiers. However, this still does not reveal any reasons behind this ubiquitous performance boost. Our next two sections address this matter.

## 5 Can Students Become Progressively Better?

### 5.1 The multi-view hypothesis

In a thought-provoking paper, Allen-Zhu & Li (2023) have proposed the "multi-view" hypothesis as a possible explanation as to why KD works so well. The multi-view hypothesis suggests that natural datasets (particularly for image classification) exhibit a special structure. Samples in such datasets consist of multiple "views" or concepts which when grouped together imply a class. For example, a car image can be correctly

classified when the model look at the headlights, the wheels, or the windows. Given a typical placement of a car in images, it is suffice to accurately predict a car using one of the above-mentioned features. The authors claim that several vision datasets (including CIFAR-10 and CIFAR-100) exhibit multi-view structure, and standard neural network models (such as ResNet-X) leverage this during training.

The authors support this hypothesis by analyzing ensembles of neural networks. They investigate how the improvement can be distilled into a single model using knowledge distillation. They then show that self-distillation is equivalent to implicitly combining ensembles and knowledge distillation to attain better test accuracy. They finally conclude that the performance boost can therefore be explained by the multi-view hypothesis. In particular, they argue that special structure in data is arguably necessary for ensemble to work. Formally, a neural network trained using the cross-entropy loss from random initialization will:

1. Learn one of the features $v \in \{v_1, v_2\}$ for the first label, and one of the features $v' \in \{v_3, v_4\}$ for the second label. As a result, 90% of the training examples consisting of features $v$ and $v'$ are classified correctly. Once classified correctly, these samples contribute negligibly to the gradient.
2. Afterwards, will memorize the remaining 10% of the training data without learning any additional features, as there is not enough data remaining after the previous phase. This explains why models can achieve 100% training accuracy but 90% test accuracy.

To elaborate, under this hypothesis, an ensemble will learn more features than a single model. Further, during knowledge distillation, the student will be forced to learn additional features from the teacher. In both cases, the resulting model will have superior test accuracy compared to an individual model trained from scratch. In the case of self-distillation, the authors suggest that the procedure implicitly combines ensemble and knowledge distillation. Particularly, if the teacher learns $\mathcal{V}_A$ features, the student is encouraged to also learn $\mathcal{V}_A$. Subsequently, it purportedly learns additional features, $\mathcal{V}_B$ on its own. Thus the self-distilled model performs better than the teacher by ensembling its independent features with those of the teacher, resulting in a larger learned set of features $\mathcal{V}_A \cup \mathcal{V}_B$.

As empirical evidence, the authors show that one-round self-distillation allows students trained on CIFAR-10/100 surpass the teacher in test accuracy. They also show that when data that does not exhibit the multi-view structure (Gaussian like with target label generated by any fully-connected / residual / convolutional network), the ensemble does not improve upon any individual model in terms of test accuracy. Lastly, they demonstrate that if we first distill knowledge from an ensemble $ens_1$ to multiple student models, and create a second ensemble $ens_2$ from those student models, then the test accuracy of $ens_2$ does not exceed $ens_1$, and is in fact lower in many cases.

If this hypothesis were true, a natural consequence would be to sequentially use self distillation to encourage student models to learn increasingly larger set of features – $\mathcal{V}_A \cup \mathcal{V}_B \cup \mathcal{V}_D$, where $\mathcal{V}_D$ are the features from model $D$ being implicitly introduced in the ensemble. We analyze if this consequence holds.

**Experiment setup.** For the first experiment, we train multiple individual models from scratch. An ensemble created from these models are then used as the teacher to perform knowledge distillation, where the student is a single model with the same architecture as the initial individual models. We increase the number of models in the ensemble from 2 to 9 and measure both the ensemble (teacher) and the student. In the next experiment, we perform self-distillation for 3 rounds. All the models have the same architecture. Each model model is trained for 600 epochs using Cutout and AutoAugment augmentations. We use 3 different $\alpha$ values of 0.2, 0.5, and 0.8. At each round, we save the model with the highest test accuracy and use it as the teacher for the next self-distillation round. An illustration of 2-round self-distillation can be seen in Figure 1. In order to remove the effect of architecture playing a role in the results, we consider a variety of architectures: Resnet-18 (He et al., 2016), VGG-16, ViT-S/32 and ViT-B/32 (Dosovitskiy et al., 2021).

**Results** We report our findings for the first experiment in Figure 3. We observe that as we increase the number of models in an ensemble and use it as the teacher, then the student will also display better test accuracy. In other words, the more features we force the student to learn, the higher test accuracy it has. If the multi-view hypothesis correctly explains self-distillation, we expect that the student learns features from

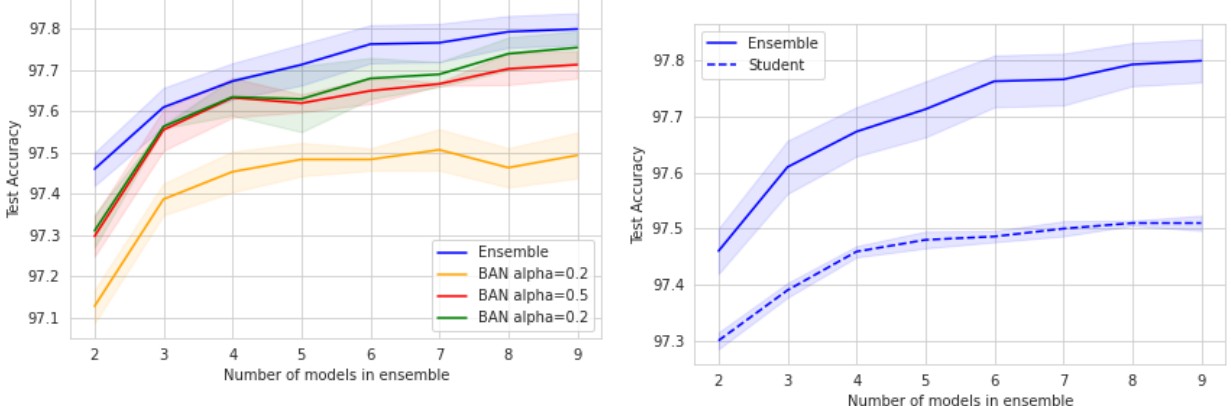

Figure 2: **BAN vs Ensemble.** *The mean and standard deviation of accuracy is reported over 3 runs. The ensemble out-performs BAN at all stages, implying that training an ensemble is more effective than multiple rounds of self-distillation.*

Figure 3: **Ensemble as teachers.** *As more models are used as teachers, the student performance improves.*

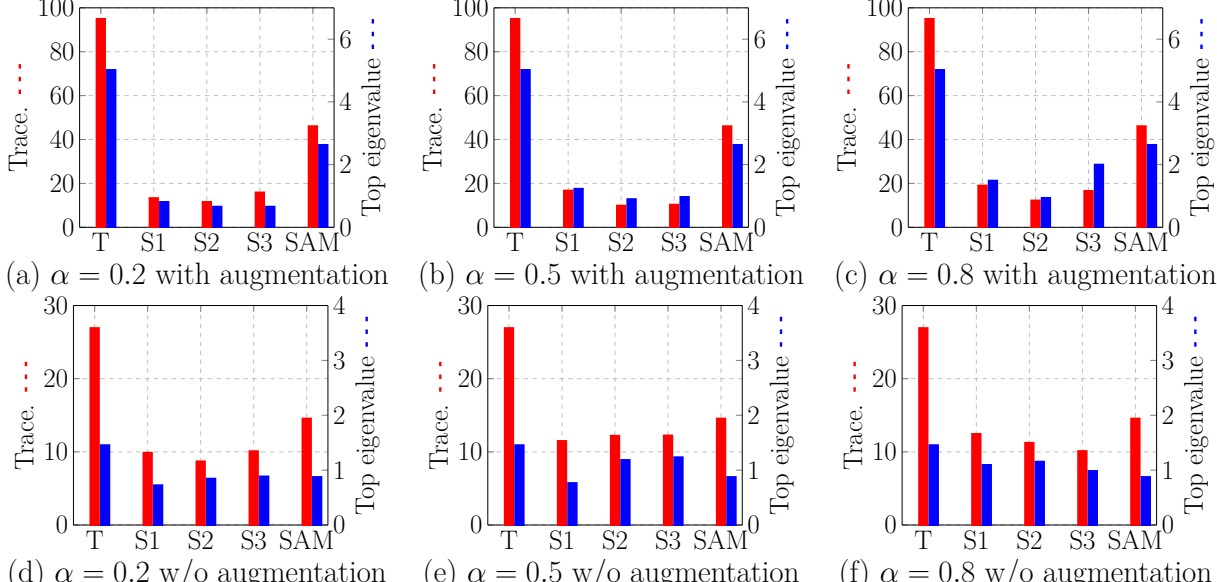

Figure 4: **Evolution of flatness measures in multi-step distillation on ResNet18 for CIFAR-10.** *'T' and 'S' stand for teacher and student models, respectively. Smaller trace (red) and $\lambda_{\max}$ (blue) values imply flatter minima. We observe the student with first round distillation enjoys getting a benefit finding flatter minima than the teacher. Surprisingly, self-distillation implicitly finds a flatter minima than SAM, which explicitly looks for the wider minima in its objective functions.*

all the previous teachers in addition to its own independent features, thus achieving incrementally better test accuracy. However, our results for the second experiment show otherwise; see Table 2. While a single round of self-distillation consistently makes the student outperform the teacher, performing it for multiple rounds does not result in a stepwise better student. For example, when the model architecture is ResNet18, performing self-distillation using $\alpha = 0.2$ without self-distillation makes the test accuracy at every step evolve as follows: $95.17\% \rightarrow 95.8\% \rightarrow 95.58\% \rightarrow 96.25\%$. We can see that the accuracy fluctuates instead of progressively increasing, which holds fold for the majority of rows in Table 2. This suggests that the multi-view hypothesis might not be sufficient to explain the success behind self-distillation.

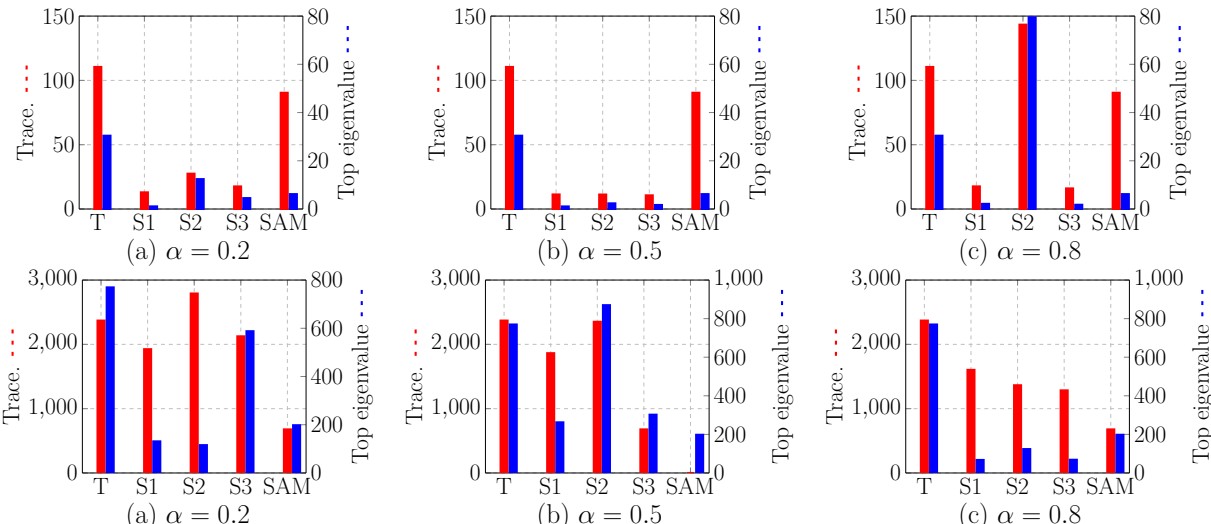

Figure 5: **_Evolution of flatness measures in multi-step distillation on VGG16 (top row) and ViT-S/32 (bottom row) for CIFAR-10._** _'T' and 'S' stand for teacher and student models, respectively. We observe the similar trends to Figure 4, which self-distillation implicitly finding wider minima than both teacher and SAM. All models use augmentation._

### 5.2 Do Born-Again Neural Networks Work?

Our proposal to perform multiple rounds of self-distillation is in fact not new, and dates back (at least) to Born-again Neural Networks (BAN) (Furlanello et al., 2018). At a high level, this involves a re-training procedure that (essentially) performs multi-round self-distillation and then constructs an ensemble of the final models of every round to make predictions. Specifically, using the notation from Figure 1, the output of the corresponding Born-Again Neural Network is given by

$$f_{BAN} = \frac{\left(f^{(0)}(x) + f^{(1)}(x) + f^{(2)}(x)\right)}{3}.$$

However, we discover that BANs actually perform worse than an ensemble over a collection of models trained independently from scratch.

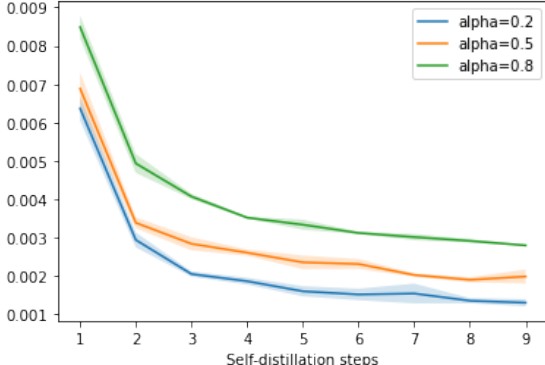

Figure 6: Difference between logits of student and its BAN teacher on CIFAR-10 train set at every self-distillation step. We calculate the discrepancy using the MSE.

**Experiment setup.** We use ResNet18 as the student models in BAN. We train the student models for 600 epochs, using SGD with momentum 0.9, weight decay $3 \times 10^{-4}$, batch size 96, gradient clipping 5.0, and an initial learning rate of 0.025. We use a cosine learning rate schedule (Loshchilov & Hutter, 2017)). For

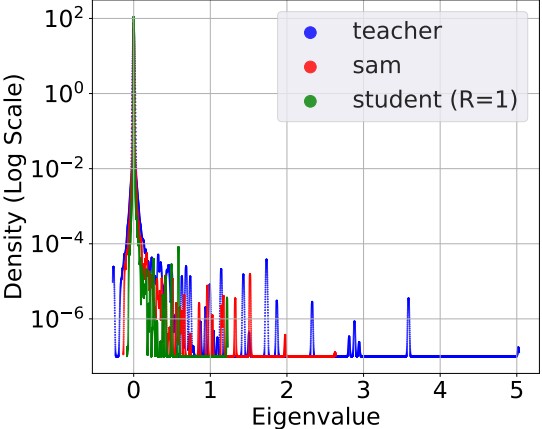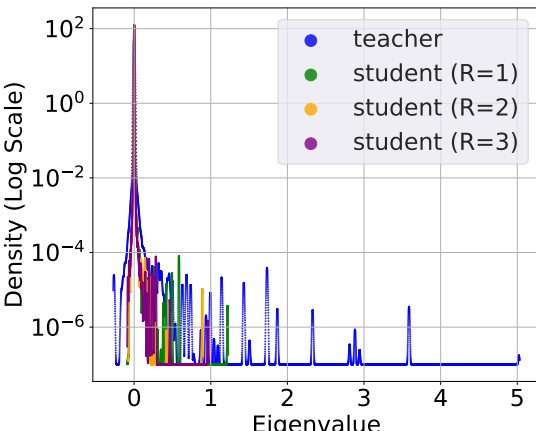

Figure 7: **Eigenspectrum of Hessian on ResNet18 from CIFAR-10** *The narrower eigenspectrum implies the flatter the loss surface. The explicit objective function from SAM (left) narrows down the eigenspectrum compared to the teacher model trained with regular cross-entropy loss. We further observe that the student model distillate from equivalent architecture (teacher) achieves an even flatter loss surface than SAM (left). The right plot compares the eigenspectrum of different students with various rounds.*

data augmentation, we also use AutoAugment (Cubuk et al. (2019)) and Cutout (Devries & Taylor (2017)). Additionally, we also train multiple models from scratch using the similar procedure to use for the ensemble. One can think of normal ensembling as a special case of BAN when $\alpha = 1.0$. We report our BAN performance on CIFAR-10.

**Results.** Figure 2 shows that BAN underperforms straightforward ensembling for all three choices of $\alpha$. Notice that as $\alpha$ increases, or as the dependency of the student on the teacher decreases, BAN performance comes closer to that of an ensemble classifier.

To investigate why BAN performs worse than standard ensembling, we calculate the difference (using Mean Squared Error) between the logits of the student and the teacher at every self-distillation step and report it in Figure 6. We can see that the more self-distillation rounds that we perform, the more similar the predictive logits of the student model and those of its teacher model. Therefore, the logits of the students will become less and less diverse. The resulting BAN model, which averages the logits of the students, will have sub-optimal performance. In other words, training BAN for multiple generations leads to initial improvements that gradually saturate, as observed by the authors, and this also indicates why increasing the number of rounds in BAN is less effective than taking an ensemble of models. This suggests that it is more effective to just simply train an ensemble from scratch than performing several rounds of self-distillation as suggested by BAN.

## 6 Why Does Self-Distillation Work?

We have empirically demonstrated that contrary to the multi-view hypothesis, multiple rounds of self-distillation fail to yield progressively better students. In this section, we propose an alternative explanation for the success of self-distillation that is more consistent with this finding.

We specifically focus on the geometry of the (local) loss landscape around the learned model parameters. The connection between landscape flatness and generalization has been extensively studied from both the empirical and theoretical perspectives (Keskar et al. (2017); Dziugaite & Roy (2017); Jiang et al. (2020); Hochreiter & Schmidhuber (1997)), and flatter minima have been reported to give better generalization in various tasks (Foret et al. (2021); Pittorino et al. (2021); Cha et al. (2021)). We will demonstrate that self-distillation makes the student model *attain flatter minima than the teacher.*

This finding is in line with previously published work: Zhang et al. (2019) have previously also hypothesized that self-distillation promotes flatter minima. However, their experiments relied on perturbing the trained weights of the teacher and the student with Gaussian noise and measuring the effect on the loss. However, due to the curse of dimensionality, a Gaussian perturbation-style analysis might not lead to accurate conclusions. In contrast, our experiments provide an alternative confirmation of their observation, but with a more direct measurement of the geometric properties of the loss landscape.

To be clear, Dinh et al. (2017) have shown that flatness on its own does not automatically imply better generalization in very deep models. Still, measuring and comparing flatness measures between the teacher and the student may provide insights on test accuracy. Similar to Chaudhari et al. (2017), we use the eigen-spectrum of the Hessian for the entire neural network to measure flatness of the loss landscape. Note that for ideal flat minima, all eigen-values of the Hessian should be positive and close to zero. This would necessarily result in also having a lower trace and lower top eigen-value $\lambda_{\max}$. We therefore also report the trace and the largest eigen-value as surrogate measures of flatness.

We use PyHessian (Yao et al. (2020)) to estimate the trace, the top eigenvalue $\lambda_{max}$, and the eigen-spectral density of the models from Table 2. PyHessian leverages standard randomized linear algebra algorithms and automatic differentiation to estimate second-order properties of large neural network models. We report the results in Figure 4 and 5. We also trained a VGG16 and a ResNet18 with the recently proposed Sharpness-Aware Minimization (SAM) (Foret et al. (2021)), an algorithm that explicitly encourages flat minima by modifying the training objective, as a suitable baseline for comparison.

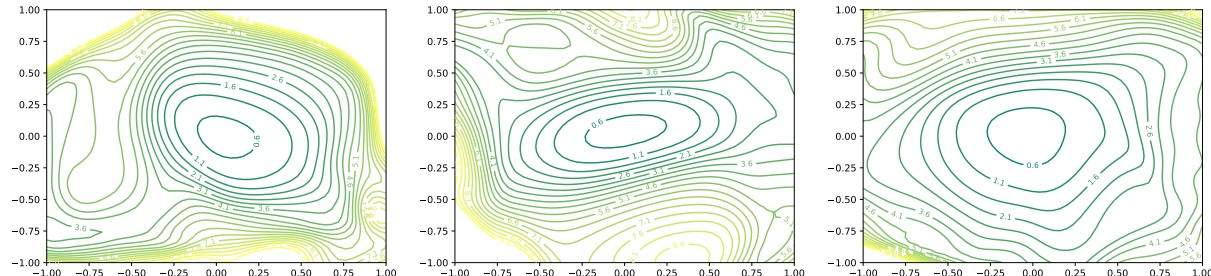

Figure 8: *Contour visualization of the loss surface (Li et al. (2018)), original teacher model (left), student at self-distillation round 1 (middle), and student at self-distillation round 2 (right). All models used ResNet20 without skip connections. The training procedure of is similar to Li et al. (2018)*
.

We notice that the teacher model trained with augmentation has a higher trace and $\lambda_{max}$ than without augmentation. Further, a single round of self-distillation will result in a student with lower trace and $\lambda_{max}$ than the teacher. Interestingly, performing multi-round self-distillation does not make successive students attain increasingly flatter minima, as the trace and $\lambda_{max}$ of models from subsequent rounds only fluctuate around those of the student from the first round. Figure 7 display the eigen-spectrum of the Hessian for ResNet18 on CIFAR-10. We can see that the overall distribution of eigenvalues of the student models is more concentrated around 0 compared to the teacher with or without SAM, therefore implying flatter minima.

Additionally, following Li et al. (2018), we trained a ResNet20 without skip connections for 2 self-distillation steps and visualize the loss surfaces similar to the authors. We demonstrate this in Figure 8. The borderlines of the teacher shows that it is much steeper than the students at both round 1 and 2. This suggests that the round-1 student achieves a flatter minima as compared to its teacher.

These observations, when combined, suggest that the self-distilled student exhibits relatively flatter minima compared to a teacher trained from scratch. This is in line with theoretically established results on induced regularization (in the context of shallow models (Mobahi et al. (2020)) and could be used to explain why (a single round of) self-distillation typically results provides test accuracy boosts.

# 7 Does Self-Distillation Give Benefits Beyond Test Accuracy?

Our above results confirm that self-distillation (even when controlling for model and dataset size) enables the student to inherit high test accuracy from the teacher. But beyond test accuracy, what other properties of the teacher does knowledge distillation transfer?

Recent works (Ojha et al., 2022; Goldblum et al., 2020) have explored this qeuestion via the lens of *robustness to distribution shifts*. In particular, Ojha et al. (2022) demonstrate that localization, adversarial vulnerability, color invariance, synthetic robustness, and biases are transferred from teacher to student through knowledge distillation. Moreover, Goldblum et al. (2020) show that an adversarially robust teacher can also make the student more robust against $\ell_\infty$ attacks. In this section, we focus on a property that has been the focus of recent study in contrastive language-image pretrained models: *natural robustness*.

**Experiment setup.** Taori et al. (2020) introduced the notion of *effective robustness* as a framework to compare the robustness of models with different accuracies. A useful tool used to study (effective) robustness are scatter plots that correlate model performance under standard settings versus distribution shift (Taori et al., 2020; Recht et al., 2019). As shown in Taori et al. (2020); Miller et al. (2021), accuracy on the reference distribution is usually an excellent predictor of accuracy under distribution shift. More formally, given models $f$, there exists a function $\beta : [0,1] \to [0,1]$ such that $Acc_{shift}(f)$ approximates $\beta(Acc_{ref}(f))$; somewhat surprisingly, for many families of models, $\beta$ is very well approximated by a straight line (Miller et al., 2021). More robust models (such as OpenAI's CLIP) have been shown to shift accuracies "above the line".

In our experiments below we measure natural robustness using classification on ImageNet (Deng et al. (2009)) as the reference, and classification on ImageNetV2 (Recht et al. (2019)) as the distribution shift.

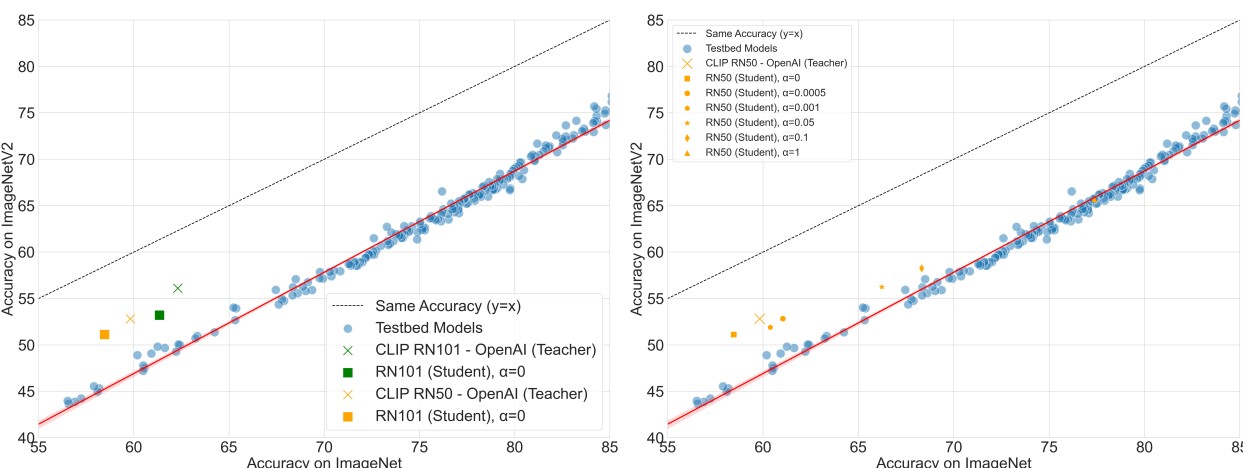

Figure 9: **Self-distillation with robust teachers.** *(Left): Self-distillation transfers effective robustness from teachers to students when $\alpha = 0$. (Right): Student models gain in-distribution accuracy but lose effective robustness quickly as $\alpha$ is increased.*

We primarily use CLIP models (Radford et al. (2021)) as teachers due to their high effective robustness. CLIP models are trained using image-caption pairs from the web. These models train an image-encoder $g$ and text-encoder $h$ such that the similarity $\langle g(x), h(t) \rangle$, where $x$ and $t$ is a pair of image and text, is maximized relatively to unaligned pairs. In order to perform zero-shot $k$-way classification, CLIP models match an image $x$ with the closest class name $c \in \{c_1, ..., c_k\}$ using potential captions. For example, using caption $t_i$ = "a photo of a $\$c_i$", for each class i, the model can make a prediction via $\arg\max_j \langle g(x), h(s_j) \rangle$. One can construct $W_{\text{zero-shot}}$ with columns $f(s_j)$ and construct $f(x) = g(x)^T W_{\text{zero-shot}}$. Specifically, we use the CLIP models with ResNet50 and ResNet101 backbones. The student models are trained using SGD using weight decay $1 \times 10^{-4}$, and batch size 1024, and an initial learning rate of 0.1 that decays every 30 epochs. During training, we only resize and center crop the images.

**Results.** Borrowing the results presented in Taori et al. (2020), we display a scatter plot of the standard versus shift accuracies of a large number of testbed (standard, supervised) image classification models in Figure 9. Visually, there is a clear linear relationship between a model's final performance on in-distribution (ID) and out-of-distribution (OOD) data. The outliers are the CLIP models (with ResNet50 and ResNet101 image backbones): they lie well above the linear fit.

Using the CLIP models as teachers, we now perform self-distillation over ImageNet, and measure the test accuracy (on ImageNet) versus shift accuracy (on ImageNetv2). We obtain interesting results. First, if we set $\alpha = 0$ in the KD loss (i.e, only retain the KL divergence term), the student models are trained to only mimic the (soft) outputs of the teacher models. Per Figure 9 (left), We observe that such student models *also* lie above the best fit line corresponding to standard testbedsm and therefore inherit the teacher's robustness; however, we also observe that the students achieve slightly worse ID and OOD performance compared to their corresponding teachers; the points have moved slightly to the left and downwards.

On the other hand, suppose we focus on the ResNet50 backbone for the teacher/student. Increasing $\alpha$ to values above zero, which introduces the use of ground-truth labels during training causes the student models significantly boosts their ID performance (up to more than 15% in base accuracy), but moves the point closer to the linear fit (and therefore loses effective robustness). Overall these experiments demonstrate that the amount of effective robustness inherited by the student models is sensitive to the choice of $\alpha$ (which influences how much guidance is provided by the teacher in the self-distillation process), and that there is a trade-off between ID and OOD performance as $\alpha$ is varied.

## 8 Discussion

In this work, we investigate several facets of self-distillation. We show that even with a strong teacher that is trained using modern techniques and augmentations, self-distillation still enables the student to surpass the teacher in terms of test accuracy. Secondly, we revisit previous literature on self-distillation and reveal potential limitations of these approaches. We then provide an alternative view on the success of self-distillation. In particular, we draw connections between self-distillation and loss geometry, and empirically show that the self-distilled student is encouraged to find flatter minima compared to the teacher; this may shed light on reasons behind its success. Finally, we show that self-distillation is able to transfer effective robustness from teachers to students, most effectively when upweighting the contribution of the teacher logits to the distillation loss.

As self-distillation is a special case of knowledge distillation (KD), we believe that understanding SD can help us develop better techniques for KD, which already has become a cornerstone of real-world state-of-the-art model building. An important open direction is the development of novel optimization procedures that implicitly perform (or emulate) self-distillation, resulting in improved student performance while avoiding cumbersome (and resource-intensive) teacher-student knowledge transfer. Moreover, another avenue of future work is to design algorithms that can achieve highly robust models without the requirement of the availability of massive datasets: dataset size has been shown to be the primary driver for robustness gains (Fang et al. (2022); Nguyen et al. (2022)), but perhaps bootstrapping with well-trained robust teachers can alleviate some of the size requirements in practical applications.

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

# A Appendix

## A Experiment Details

For training the neural networks, we use SGD with momentum of 0.9, learning rate 0.025, weight decay $3 \times 10^{-4}$, batch-size 96, and gradient clipping value of 5.0.

## B Additional experiments on CIFAR-10/100

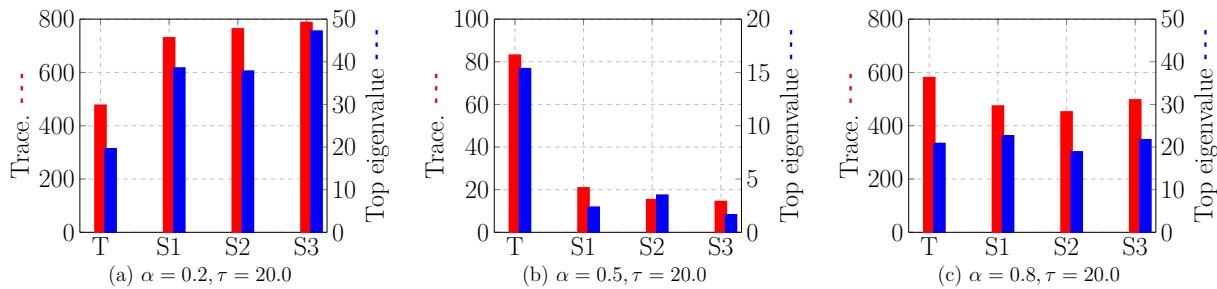

Figure 10: Tracking trace and top eigenvalue in distillation steps on VGG16 for CIFAR100. All models use augmentation.

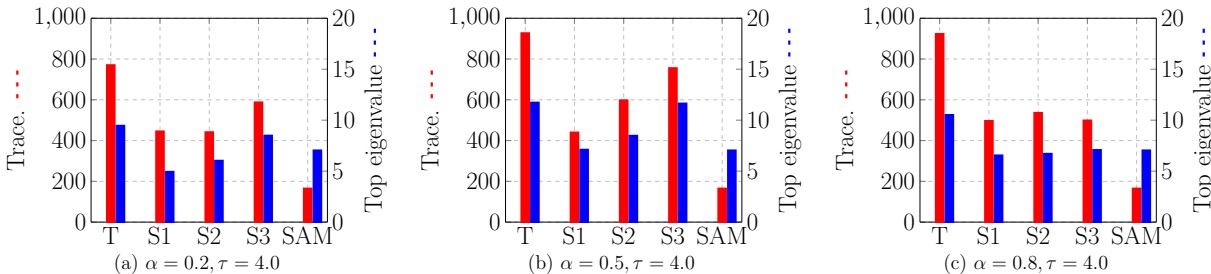

Figure 11: Tracking trace and top eigenvalue in distillation steps on VGG16 for CIFAR10. All models use augmentation.

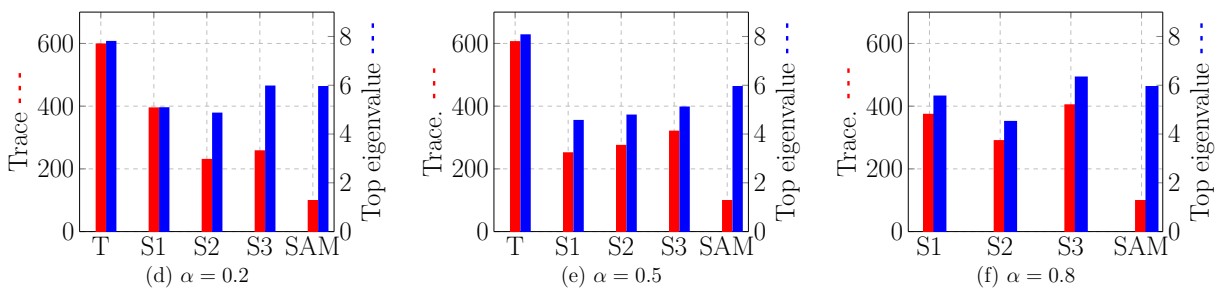

Figure 12: *Tracking trace and top eigenvalue in distillation steps on ResNet18 (bottom row) for CIFAR-100. All models use augmentation.*

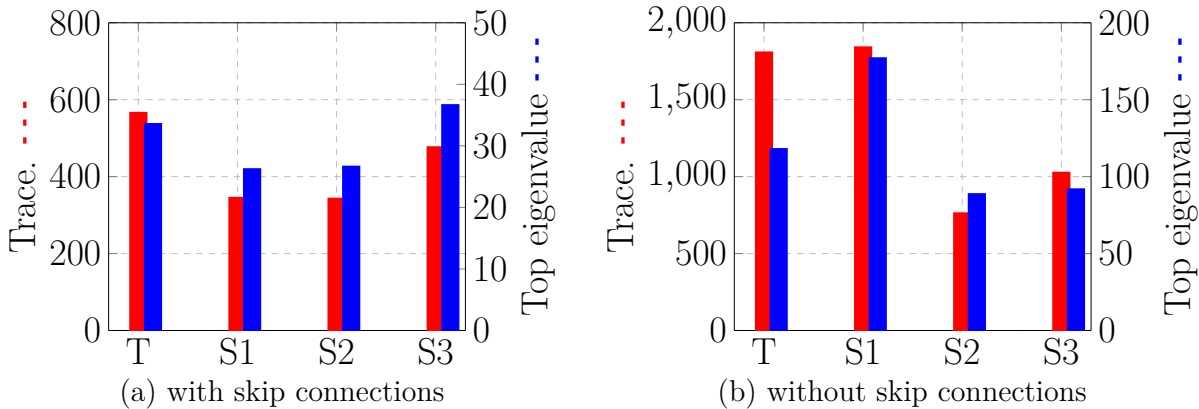

(a) with skip connections  (b) without skip connections

Figure 13: *Tracking trace and top eigenvalue in distillation steps on ResNet20 for CIFAR-10. All models use augmentation.*

| Architecture | Dataset | Data Aug. | $\alpha$ | Teacher | Round 1 | Round 2 | Round 3 | SAM |
|---|---|---|---|---|---|---|---|---|
| ResNet20 w/o skip connections | CIFAR-10 | Yes | 0.2 | $93.28 \pm 0.12$ | $93.5 \pm 0.04(\uparrow)$ | $93.42 \pm 0.11(\downarrow)$ | $93.61 \pm 0.03(\uparrow)$ | $93.52 \pm 0.04$ |
| ResNet20 w/o skip connections | CIFAR-10 | No | 0.2 | $92.94 \pm 0.10$ | $93.38 \pm 0.03(\uparrow)$ | $93.52 \pm 0.14(\uparrow)$ | $93.48 \pm 0.03(\downarrow)$ | $93.50 \pm 0.04$ |
| ResNet20 | CIFAR-10 | Yes | 0.2 | $93.25 \pm 0.20$ | $93.99 \pm 0.15(\uparrow)$ | $93.75 \pm 0.10(\downarrow)$ | $93.17 \pm 0.58(\downarrow)$ | $94.01 \pm 0.07$ |
| ResNet20 | CIFAR-10 | No | 0.2 | $92.69 \pm 0.06$ | $93.46 \pm 0.12(\uparrow)$ | $93.48 \pm 0.04(\uparrow)$ | $93.60 \pm 0.18(\uparrow)$ | $93.51 \pm 0.10$ |

Table 3: **Self-distillation results on CIFAR-10.** *Data augmentation means leveraging Cutout and AutoAugment techniques. We report mean and standard deviations of test accuracy from three independent runs.* $\uparrow$ *(resp.* $\downarrow$*) stands for the increase (resp. decrease) in test accuracy relative to its teacher.*

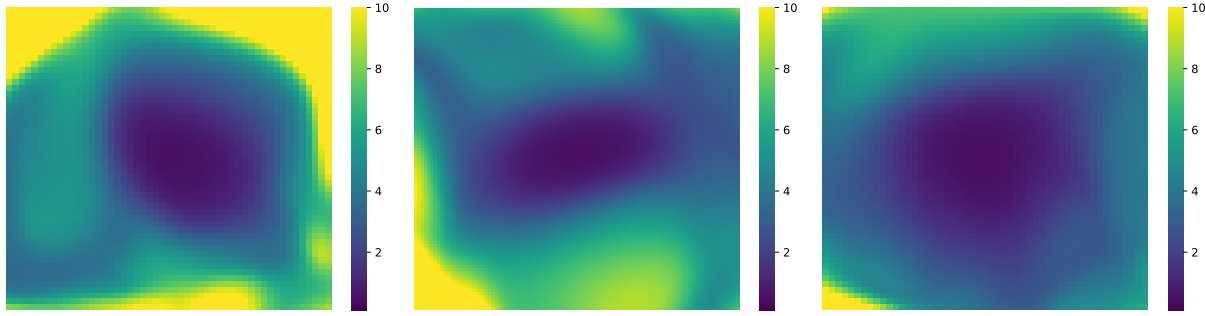

Figure 14: *2D heatmap of the loss surface (Li et al. (2018)), original teacher model (left), student at self-distillation round 1 (middle), and student at self-distillation round 2 (right). All models used ResNet20 without skip connections. The training procedure of is similar to Li et al. (2018)*
.

## C  SVHN results

|  | Accuracy | Trace | $\lambda_{\max}$ |
|---|---|---|---|
| Teacher | 95.23 | 197.53 | 9.24 |
| Round 1 ($\alpha = 0.5$) | 95.94 | 205.79 | 11.200 |
| Round 2 ($\alpha = 0.5$) | 95.67 | 98.62 | 8.30 |
| Round 3 ($\alpha = 0.5$) | 95.17 | 271.71 | 12.873 |

Table 4: SVHN results for ResNet18

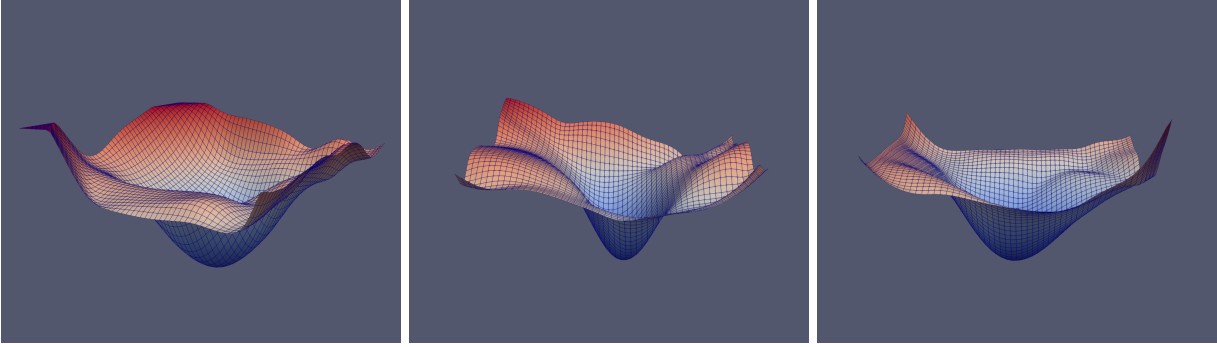

Figure 15: *3D visualizations of the loss surface (Li et al. (2018)), original teacher model (left), student at self-distillation round 1 (middle), and student at self-distillation round 2 (right). All models used ResNet20 without skip connections. The training procedure of is similar to Li et al. (2018)*
.

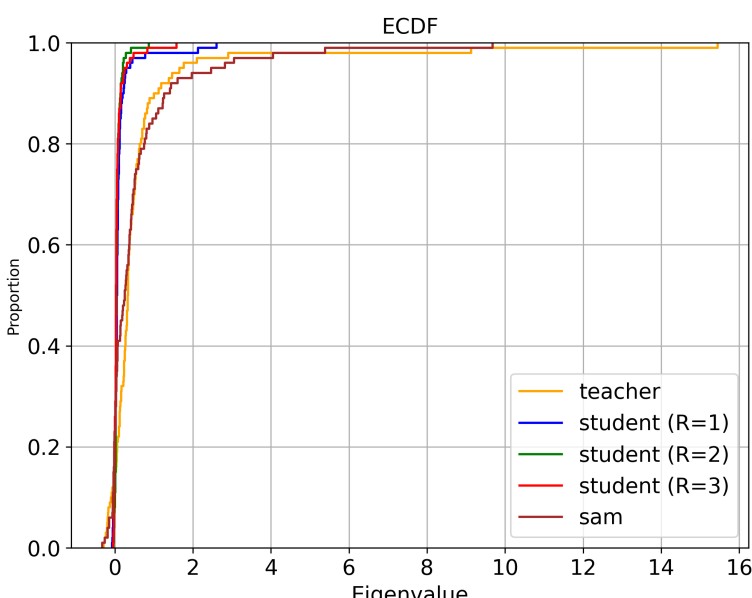

Figure 16: ECDF plot of the top 100 eigenvalues approximated using PyHessian. All models are ResNet-18 trained on CIFAR-10 with data augmentations.

## D  Tiny Imagenet results

|  | Accuracy | Trace | $\lambda_{\max}$ |
|---|---|---|---|
| Teacher | 62.49 | 415.64 | 13.52 |
| Round 1 ($\alpha = 0.2$) | 65.45 | 352.12 | 12.10 |
| Round 2 ($\alpha = 0.2$) | 66.26 | 361.34 | 14.23 |
| Round 3 ($\alpha = 0.2$) | 65.99 | 332.67 | 13.45 |
| SAM | 63.26 | 310.87 | 8.04 |

Table 5: Tiny ImagenetNet results for ResNet18. We leverage random cropping and random rotation as data augmentations.

