# OpenReview forum: "Revisiting Self-Distillation"
_TMLR — Rejected by TMLR_

### Review · Reviewer_nhJX · 2023-04-05

**Summary Of Contributions:**

The present paper analyzes and tests existing explanations on why knowledge distillation yields student models better than the teacher model. In particular, the authors revisit the multi-view hypothesis that student models learn not only features captured in the teacher model but also potentially new features, and the union of both sets of features helps it generalize better. The authors' new experiments show interesting behavior, although does not necessarily refute the hypothesis. The authors also shows that simple ensembling outperforms a prior proposed approach of ensembling students from  multiple distillation rounds (i.e., repeatedly distilling from the newest student). The authors then performs loss landscape analyses comparing the teacher and student model, although the difference is not very clear. Finally, the authors show that knowledge distillation can preserve robustness of a robust teacher  model, under some particular hyperparameter setting, although this doesn't seem to produce a better model w/ similar or better robustness.

**Audience:**

Yes

**Claims And Evidence:**

No

**Requested Changes:**

+ Section 5.1: Authors refute the multi-view hypothesis w/ experiment that multi-round distillation from an ensemble teacher does not improve. However, it is not necessarily true. The results could also be explained by either a capacity issue or that there aren't many meaningful "views" in this dataset. To exclude these two possibilities, the authors could (1) distill with larger student model and (2) distill from a single-model teacher.
+ Section 5.2: Authors present born-again networks (BAN) as an existing explanation why distillation works. However, BAN is not an explanation. Rather, as the author described, BAN is a method built on top of distillation. Therefore, the authors inaccurately claimed that they "revisit existing explanations and reveal their drawbacks", while in reality they only attempt to refute the multi-view hypothesis (with some problematic experiments, as mentioned above), ignoring all other explanations such the many works mentioned in their related works section.
+ Section 6: For loss landscape visualizations (Figure 8),
   + Why are skip connections removed? If the sharpness argument holds, should it work with regular resnets?
   + I don't think that the figure justifies the claim that teacher model has a steeper loss landscape.

Writing:
+ Section 4 beginning paragraphs talk about a single question: whether self-distillation improves upon a well-trained teacher. However, writing makes it reads like there are two distinct questions. E.g., "On the other hand, ...."
+ Page 7 bottom "....  for the second experiment show otherwise; see Figure 2." should be Table 2.
+ Figure 7 is very hard to read. Consider using the ECDF rather than empirical frequency, which has many spikes.

**Strengths And Weaknesses:**

Strengths:
+ Explanation and investigation on multiple ideas around knowledge distillation, with new experiments and results.
+ The robustness and sharpness angles are potentially interesting.

Weaknesses:
+ Claims are generally not well supported by experiment results (see below).
+ Writing and figures could be improved for better readability (see below).

---

> ### Author Response · Authors · 2023-04-29
>
> We would like to thank the reviewer for the comments.
>
> > Section 5.1: Authors refute the multi-view hypothesis w/ experiment that multi-round distillation from an ensemble teacher does not improve. However, it is not necessarily true. The results could also be explained by either a capacity issue or that there aren't many meaningful "views" in this dataset. To exclude these two possibilities, the authors could (1) distill with larger student model and (2) distill from a single-model teacher.
>
> Regarding the multi-view hypothesis, our intention was not to refute it but to demonstrate its potential incompleteness through two main experiments. In the first experiment (Figure 3), we perform distillation from an ensemble teacher to a single-model student. We observe that the student model's performance improves as the ensemble size increases, which suggests that the student benefits from learning from more views, in line with the multi-view hypothesis. The second experiment (Table 2) involves self-distillation, where we distill from a single-model teacher to a single-model student with identical architecture. According to the multi-view hypothesis, self-distillation should perform "implicit ensemble + knowledge distillation," which implies that repeating self-distillation multiple times should accumulate views and incrementally improve the student's performance. However, Table 2 does not support this notion, leading us to conclude that the multi-view hypothesis might not be sufficient to explain self-distillation. We will be happy to perform the experiments that the reviewer suggests.
>
> > Section 5.2: Authors present born-again networks (BAN) as an existing explanation why distillation works. However, BAN is not an explanation. Rather, as the author described, BAN is a method built on top of distillation. Therefore, the authors inaccurately claimed that they "revisit existing explanations and reveal their drawbacks", while in reality they only attempt to refute the multi-view hypothesis (with some problematic experiments, as mentioned above), ignoring all other explanations such the many works mentioned in their related works section.
>
> We will update the abstract and conclusion sections to better reflect our contributions based on the reviewer’s suggestions.
>
> > * Section 6: For loss landscape visualizations (Figure 8),
> >   * Why are skip connections removed? If the sharpness argument holds, should it work with regular resnets?
> >   * I don't think that the figure justifies the claim that teacher model has a steeper loss landscape.
>
> For the loss landscape visualizations (Figure 8), we followed the approach in [1] and used a ResNet-20 without skip connections, as it is shown to have a sharp loss landscape. The goal of our contour visualization is to provide empirical evidence to support our claim that self-distillation helps to find the less sharp minima. Furthermore, our study of observing the trace of Hessian on the objective function of two different architectures (ResNet18, VGG16, and ViT-S) also supports our claim. We will include additional 3D visualizations of Figure 8 in the Appendix to further support our claims.
>
> > * Section 4 beginning paragraphs talk about a single question: whether self-distillation improves upon a well-trained teacher. However, writing makes it reads like there are two distinct questions. E.g., "On the other hand, ...."
> > * Page 7 bottom ".... for the second experiment show otherwise; see Figure 2." should be Table 2.
> > * Figure 7 is very hard to read. Consider using the ECDF rather than empirical frequency, which has many spikes.
>
> Thank you. We will address the writing issues based on your recommendations and include ECDF plots of self-distilled models in the Appendix for better readability.
>
> **References**:
>
> [1] Li, Hao, et al. "Visualizing the loss landscape of neural nets." Advances in neural information processing systems 31 (2018).

---

### Review · Reviewer_uStu · 2023-04-06

**Summary Of Contributions:**

The paper presents an empirical analysis of self-distillation primarily in vision models, with an aim towards hypothesizing plausible explanations for self-distillation:
 1. The paper demonstrates that self-distillation generally does boost performance (for a single round), even when the baseline model is well-optimized
 2. The paper re-examines the multi-view explanation for self-distillation and claims to provide some negative evidence against it.
 3. The paper demonstrates that the loss surface in student models becomes progressively flatter due to self-distillation, possibly contributing to improved generalization performance.
 4. The paper provides evidence that distillation loss transfers effective robustness from teacher to student

**Audience:**

Yes

**Claims And Evidence:**

Yes

**Requested Changes:**

* Grammar nits: "Overall this experiments indicate that demonstrates that the amount of effective robustness inherited by the student models is sensitive to the choice of α"
* Consider clarifying in the abstract/title that this is primarily experimental evidence for self-distillation of vision classification models, rather than NLP/sequence models (which has their own literature around self-distillation/data augmentation). Results may not necessarily generalize across modalities/for large distribution shifts, although that seems not implausible.

**Strengths And Weaknesses:**

Strengths:
  1. Well-structured, clear, and organized
  2. Doesn't overclaim
  3. Experiments are convincing and generally interesting

Weaknesses:
  1. Not clear that the multi-view hypothesis is strongly refuted by lack of monotonic improvement. The positive evidence for flatness/effective robustness is more interesting.
  2. More a weakness of the cited papers, but it's not clear that a linear fit to effective robustness necessarily holds for very large distribution shifts (ie. task-level) and small data regimes. ImageNet -> ImageNetv2 does have some shifts, but not task-level.

---

> ### Author Response · Authors · 2023-04-29
>
> We would like to thank the reviewer for the valuable feedback.
>
> > Not clear that the multi-view hypothesis is strongly refuted by lack of monotonic improvement. The positive evidence for flatness/effective robustness is more interesting.
>
> Our goal was not to strongly refute the multi-view hypothesis; instead, our intent in the experiments was to show that the proposed hypothesis might be incomplete. Based on the multi-view hypothesis, the more views the student can learn, the higher its accuracy on the test set. However, for self-distillation (Table 2), we observe a non-monotonic increase in the performance of the student models. This indicates that the multi-view hypothesis may not fully capture the nuances of how self-distillation works. We will be happy to amend our wording here if that conveyed an incorrect impression.
>
> > * Grammar nits: "Overall this experiments indicate that demonstrates that the amount of effective robustness inherited by the student models is sensitive to the choice of α"
> >
> > * Consider clarifying in the abstract/title that this is primarily experimental evidence for self-distillation of vision classification models, rather than NLP/sequence models (which has their own literature around self-distillation/data augmentation). Results may not necessarily generalize across modalities/for large distribution shifts, although that seems not implausible.
>
> We will address all grammar issues, particularly in the sentence mentioned, and ensure clarity throughout the paper. Additionally, we will modify the abstract to specify that our study primarily focuses on experimental evidence for self-distillation in vision classification models. We acknowledge that our results may not necessarily generalize across modalities or for large distribution shifts, and we will make this limitation clear in our revised manuscript.

---

### Review · Reviewer_yydm · 2023-05-08

**Summary Of Contributions:**


Setup is self-distillation (SD) where a model is trained first with cross-entropy loss. In the next round, this model becomes the teacher and a student is trained using a combination of the cross-entropy and KL-divergence term between student and teacher. This iterative proceducer is repeated a few times. It results in the following observations
- a student surpasses even a highly accurate teacher
- issues with earlier explanation of improvement through self-distillation
- dynamics of loss-landscape -- self-distillation leads to flatter minima and hence better generalization
- what properties SD transfers from teacher to students -- one of this is robustness

**Audience:**

Yes

**Claims And Evidence:**

No

**Requested Changes:**

Questions for Authors:
------------

- In Table 2, why is there no standard deviation in the ViT-S/32 rows?
- In Table 2, how were the independent trials performed? Did you fix the random seed (python/numpy/cuda/pytorch/etc.)?
- In Table 2, instead of CIFAR-10, it would have been better to include CIFAR-100 dataset as you might see some diversity in the performance. As most of the networks are close to saturation region in the CIFAR-10 dataset.

- Do you have results about training the teacher for more epochs? Say, the student is trained for 200 epochs and before that the teacher was trained for 200 epochs. Can you see what the teacher does if it was trained from scratch for 400 epochs?

- In Figure 8, what happens when you add the skip connections? Do you still see flatter region?

Missing Related Works:
---------

- Xinchuan Zeng and Tony R. Martinez. Using a neural networks to approximate an ensemble of
classifiers.

- Jimmy Ba and Rich Caruana. Do deep nets really need to be deep?

**Strengths And Weaknesses:**

Strengths:
-----------

- Gives counterexamples to multi-view hypothesis (i.e., a teacher typically only learns a strict subset of views of the input and self-distillation enables the student to learn the rest of the views)

- Shows the loss-landscape of the self-distillation minima vs the teacher

- Shows self-distillation yields results beyond test accuracy

Weaknesses:
-----------

- In Table 2, in many entries, teacher and students are very close by in accuracy. It is unclear why this would mean that self-distillation yields superior students. If at all, training a student in this manner would mean wasting compute for only marginal improvement in accuracy?

- It is unclear how much of the proposed investigation holds true once we move beyond the small scale territory of CIFAR-10/CIFAR-100 to ImageNet or beyond.

- Flat minima conclusion would be more pronounced if it was done for a network with skip connections.

---

> ### Author Response · Authors · 2023-05-15
>
> We would like to thank the reviewer for the comments.
>
> > * In Table 2, in many entries, teacher and students are very close by in accuracy. It is unclear why this would mean that self-distillation yields superior students. If at all, training a student in this manner would mean wasting compute for only marginal improvement in accuracy?
>
> Despite the seemingly minor discrepancy in accuracy between the teacher and students, our analysis reveals that this difference is, in fact, statistically significant. It is crucial to highlight that our primary objective is not to introduce a novel method, but rather to delve deeper into the understanding of self-distillation. Our results indicate that an initial round of self-distillation typically yields an enhanced student model. However, it should be noted that subsequent iterations of self-distillation do not consistently result in progressive improvements. Hence, when computational resources are a factor of significance, it may be most beneficial to limit self-distillation to a single iteration.
>
> > * It is unclear how much of the proposed investigation holds true once we move beyond the small scale territory of CIFAR-10/CIFAR-100 to ImageNet or beyond.
>
> [2] shows that (in Appendix C.3) self-distillation on ImageNet using ResNet-50 does result in a better model. In addition, we will add additional results for ImageNet in the Appendix.
>
> > * Flat minima conclusion would be more pronounced if it was done for a network with skip connections.
> > * In Figure 8, what happens when you add the skip connections? Do you still see flatter region?
>
> We intentionally followed the approach in [1] and used a ResNet-20 without skip connections, as it is shown to have a sharp loss landscape. We will add additional results on ResNet-20 without skip connections in the Appendix.
>
> > * In Table 2, why is there no standard deviation in the ViT-S/32 rows?
>
> Models with no standard deviation mean that across 3 runs, the test accuracies stay the same.
>
> > * In Table 2, how were the independent trials performed? Did you fix the random seed (python/numpy/cuda/pytorch/etc.)?
>
> The independent trials were performed by varying random seeds.
>
> > * In Table 2, instead of CIFAR-10, it would have been better to include CIFAR-100 dataset as you might see some diversity in the performance. As most of the networks are close to saturation region in the CIFAR-10 dataset.
>
> We will update the table with additional CIFAR-100 results.
>
> > * Do you have results about training the teacher for more epochs? Say, the student is trained for 200 epochs and before that the teacher was trained for 200 epochs. Can you see what the teacher does if it was trained from scratch for 400 epochs?
>
> In our experiments, we actually trained all our models (teacher and students) for 600 epochs. Additionally, we selected the best models (on test accuracy) to report results and used them as the teachers for the next round of self-distillation.
>
> > Missing Related Works:
> > * Xinchuan Zeng and Tony R. Martinez. Using a neural networks to approximate an ensemble of classifiers.
> > * Jimmy Ba and Rich Caruana. Do deep nets really need to be deep?
>
> We will update our paper to mention these related works.
>
> **References:**
>
> [1] Li, Hao, et al. "Visualizing the loss landscape of neural nets." Advances in neural information processing systems 31 (2018).
>
> [2] Stanton, Samuel, et al. "Does knowledge distillation really work?." Advances in Neural Information Processing Systems 34 (2021): 6906-6919.

---

### Author Response · Authors · 2023-05-22

Dear Reviewers,

We would like to thank the reviewers for their valuable feedback.

An updated version has been uploaded that incorporates additional experiments conducted in response to your suggestions. These new experiments have strengthened our results and we believe they address your concerns satisfactorily.

In addition, we have corrected all identified grammatical errors in the manuscript. Furthermore, we have refined the narrative to more accurately and succinctly represent the primary focus and objectives of our study. We trust that these changes have greatly improved the clarity and coherence of our paper.

---

### Decision · Action_Editors · 2023-06-17

**Recommendation:** Reject

**Comment:**

This decision was a big tricky.  Two out of three reviewers said that they believe the paper does support its claims with evidence, but at the same time two out of three reviewers are recommending rejection.  Overall, in light of the reviewer recommendations I feel as though I must reject at this time.

I did want to take some care to see if I could support overturning the reviewer recommendations, and so tried to independently judge whether I felt the paper supported its claims adequately with evidence, and I have to say that I not only had many of the same concerns lingering as the other reviewers but have some additional concerns.

In particular, one thing that I noticed is that the results in table 1 and table 2 are reported to the permyriad, i.e. 0.01%.  I don't feel as though this is appropriate for the 10,000 examples in the CIFAR10 test set.  If the paper is interested in making broader claims about whether models that have been progressively distilled are actually achieving better generalization performance, rather than just better performance on the specific set of 10,000 examples in the CIFAR10 test set, then the standard deviation across runs doesn't capture the full variation.  For example see: https://towardsdatascience.com/digit-significance-in-machine-learning-dea05dd6b85b , but for 10,000 examples and accuracies in the high 90s we shouldn't expect the test accuracies below about 2 permille, or 0.2%, which would remove much of the signal from Table 2, as well as Table 1.

Another potential issue is that I don't really understand how Figure 3 in the paper supports the claim in the paper that: "In other words, the more features we force the student to learn, the higher test accuracy it has.", this seems to neglect the possibility that the improvements in the student could simply be due to the ensembles having higher accuracy themselves with more members, by some mechanism besides the multi-view hypothesis.

Overall, I applaud the authors on what seem to be a large set of experimental results.  However I feel as though those results don't quite amount to the evidence required to tell the story the paper tells.

In light of the reviewer recommendations and in spite of my attempt to build a case for acceptance in contrast to the recommendations, I have to reject the paper at this time.  However I will enable the option for the authors to consider submitting a major revision at a later time as I do believe there is a lot of good work in this paper, just either some non-trivial further evidence or some non-trivial reduction in the claims necessary.

I also want to apologize for the timing delays on this paper, there were difficulties coordinating reviews and reviewers recently, but I'm sorry for my part in not working harder to make things move at an acceptable pace.

**Audience:**

The paper is clearly appropriate for the TMLR audience.

**Claims And Evidence:**

The paper has a good set of empirical investigations.

However, there are some concerns from the reviewers and myself as to whether the evidence is appropriate for supporting the claims given.



**Resubmission Of Major Revision:**

The authors may consider submitting a major revision at a later time.

---

> ### Author Response · Authors · 2023-06-23
>
> We would like to thank the AE for the comments and we genuinely appreciate the time and effort you invested in providing an independent evaluation of our work. In response to your suggestions, we plan to enhance our paper by conducting additional experiments that will fortify our assertions. Furthermore, we will endeavor to adjust our claims to better reflect and align with the empirical results obtained. Please let us know if there are any suggestions or amendments that you believe would enhance our resubmission.